# A Quassinoid Diterpenoid Eurycomanone from *Eurycoma longifolia* Jack Exerts Anti-Cancer Effect through Autophagy Inhibition

**DOI:** 10.3390/molecules27144398

**Published:** 2022-07-08

**Authors:** Guiqin Ye, Mengting Xu, Yuhan Shu, Xin Sun, Yuanyuan Mai, Yupeng Hong, Jianbin Zhang, Jingkui Tian

**Affiliations:** 1Hangzhou Medical College, Hangzhou 310014, China; 88101202033@hmc.edu.cn (G.Y.); 881012021036@hmc.edu.cn (Y.M.); 2Cancer Center, Department of Medical Oncology, Zhejiang Provincial People’s Hospital (Affiliated People’s Hospital, Hangzhou Medical College), Hangzhou 310014, China; 21815087@zju.edu.cn (M.X.); 11615018@zju.edu.cn (Y.S.); sunxin@hmc.edu.cn (X.S.); hypbdn@zju.edu.cn (Y.H.); 3College of Biomedical Engineering and Instrument Science, Zhejiang University, Hangzhou 310058, China; 4The Cancer Hospital of the University of Chinese Academy of Sciences (Zhejiang Cancer Hospital), Institute of Basic Medicine and Cancer (IBMC), Chinese Academy of Sciences, Hangzhou 310002, China

**Keywords:** eurycomanone, autophagy, angiogenesis, mTOR, colon cancer

## Abstract

Eurycomanone (EN) is one of the representative quassinoid diterpenoids from roots of *Eurycoma longifolia* Jack, a natural medicine that is widely distributed in Southeast Asia. Previous studies showed that EN induces cancer cell apoptosis and exhibits anti-cancer activity, but the molecular mechanism of EN against cancer has still not been elucidated. In this study, we examined the regulatory effect of EN on autophagy to reveal the mechanism of EN-mediated colon cancer growth inhibition. First, we found that EN is able to inhibit colon cancer cell proliferation and colony formation. The angiogenesis level in cancer cells was inhibited as well. Next, the treatment of EN led to the suppression of autophagy, which was characterized by the downregulation of the LC3-II level and the formation of GFP-LC3 puncta under EN treatment in colon cancer. Moreover, we revealed that the mTOR signaling pathway was activated by EN in a time- and concentration-dependent manner. Finally, autophagy induction protected colon cancer cells from EN treatment, suggesting that autophagy improves cell survival. Taken together, our findings revealed the mechanism of EN against colon cancer through inhibiting autophagy and angiogenesis in colon cancer, supporting that the autophagy inhibitor EN could be developed to be a novel anti-cancer agent.

## 1. Introduction

Global statistics showed that colon cancer was the third most common cancer with the second highest mortality among all cancers [1]. Experts predicted 53,200 deaths from colon cancer during 2020 in America [2]. Compared with European and American countries, the incidence of colon cancer in China is higher and has shown a clear upward trend in recent years [3]. Colon cancer occurs in the colon or rectum and is associated with high-fat and low-fiber diets. Due to irregular eating schedules or unhealthy eating habits, the age of onset of colon cancer is decreasing, and the incidence of cancer is rising among people in their 20s and 30s. Thus, it is urgent to develop effective treatments for colon cancer. Nowadays, there are three common clinical treatment methods, including surgery, chemotherapy, and radiotherapy, among which surgery is still the most effective choice in the early stage. However, for advanced colon cancer, combined treatments must be applied. Although chemotherapy agents such as 5-fluorouracil, capecitabine, irinotecan, and oxaliplatin [4,5,6,7] can be effectively used alone or in combination to treat colon cancer, their side effects and secondary drug resistance are problems that we cannot ignore. For the past few years, targeted drugs, such as the angiogenesis inhibitors bevacizumab [8] and ramucirumab [9], have been successfully applied in the treatment of advanced colon cancer. Unfortunately, some serious adverse effects accompany cancer therapy, such as hypertension, hemorrhage, etc. Therefore, it seems to be necessary to develop anti-colon-cancer drugs with fewer side effects at the current moment.

As the potent anti-neoplastic activity of paclitaxel has been observed [10], more and more studies focus on the pharmacodynamics of natural products with less toxic side effects and broad prospects for anti-cancer therapies [11]. Eurycomanone (EN) is extracted from the root of *Eurycoma longifolia* Jack, which is widely distributed in southeast Asian countries. The traditional usages and toxicity of *Eurycoma longifolia* Jack have been reviewed [12,13,14], and it was revealed to possess good anti-angiogenesis and anti-tumor effects and to have the potential to be a new anti-cancer drug. Among the variety of components of *Eurycoma longifolia* Jack, quassinoid diterpenoids are the most effective and have more than three valuable pharmacological effects, such as improved sexual function, enhanced male fertility, anti-malarial activity, and anti-tumor activity. EN is one of the representative quassinoid diterpenoids and exhibits anti-proliferation and cytotoxicity in different cancers [15,16] such as breast cancer and lung cancer. Here, our group extracted the components of *Eurycoma longifolia* Jack and further improved the extraction and purification process of EN (Figure 1A). In addition, the inhibitory effect of EN on colon cancer was further explored in our study. 

Autophagy is an ancient biological process that is prevalent in eukaryotic cells. The process of autophagy can be simply divided into three steps: the two-layer membrane structure wrapping the components to be degraded to form autophagosomes, the fusion of autophagosomes with lysosomes to form autolysosomes, and the degradation of the autophagic components [17]. Autophagy plays a crucial role in multiple physiological processes [18], such as aging, development, cell death and survival, etc. Most importantly, autophagy has been closely associated with tumorigenesis [18,19], either suppressing tumors or promoting tumors. In the treatment of cancer, many anti-cancer agents exert their function through regulating autophagy, either inducing autophagy [20,21] or inhibiting autophagy [22,23]. Whether the tumor-suppressive effect of EN is associated with autophagy is not known. 

Angiogenesis refers to the development of new blood vessels from pre-existing capillaries or capillary veins. It plays a fundamental role in the occurrence, progression, and migration of tumors through the transportation of nutrients and oxygen [24]. Currently, more and more studies have proven that the inhibitors of angiogenesis can be used for cancer therapy. They were designed to target VEGF, the VEGF receptor, or other specific molecules instead of acting directly on cancer cells [25,26,27]. Similarly, drug resistance occurs with the use of anti-angiogenic therapies [28], which are associated with autophagy. Thus, it is possible to overcome drug resistance and enhance the efficacy of angiogenesis inhibitors through regulating the autophagy level in cancer. 

Here, we hypothesized that EN exerts an anti-colon-cancer effect by regulating autophagy. In this study, our results showed that EN inhibits colon cancer cell growth in a time- and dose-dependent manner. Under EN treatment, the autophagy level of cancer cells decreased, which can be attributed to the activation of the mTOR signaling pathway. Functionally, autophagy enhancement plays a protective role in the EN-induced cell growth inhibition of colon cancer. Taken together, our findings demonstrate that EN might be a novel therapeutic agent to use alone or in combination with autophagy inhibitors in the treatment of colon cancer.

## 2. Materials and Methods

### 2.1. Reagents and Antibodies

The reagents used in our research were: a BCA protein assay kit (Solarbio, #PC0020), BeyoECL Plus (Beyotime, #P0018S, Shanghai, China), CQ (Sigma Aldrich, #C6628, St. Louis, MO, USA), crystal violet (Beyotime, #C0121), DCFH-DA (Beyotime, #S0033S), DMSO (Solarbio, #30072418, Beijing, China), EBSS (Gibco, #24010043, Waltham, UK), GSH (AAT, #22810, Sunnyvale, California, USA), a Hoechst33342 staining kit (Beyotime, #C1025), Lipofectamine 2000 reagent (Invitrogen, #11668027), matrigel (BD, #356234, New York, USA), MTT (Solarbio, #M8180, Beijing, China), a CCK-8 kit (Yeasen,#40203ES60, Shanghai, China), Opti-MEM (Gibco, #11095080), and RAPA (Beyotime, #S1842).

The antibodies used in the experiments were from Cell Signaling Technology (CST): α-Tubulin (CST, #2144), β-actin (CST, #58169), Beclin-1 (CST, #3495), ERK (CST, #4695), GAPDH (CST, #2118), LC3 (CST, #3868), mTOR (CST, #2983), phospho-ERK (Thr202/Tyr204, CST, #4370), phospho-mTOR (Ser2448, CST, #5536), phospho-S6 (Ser235/236, CST, #2211), phospho-VEGFR2 (Tyr1175, CST, #2478), S6 (CST, #2217), TSC2 (CST, #4308), and VEGFR2 (CST, #2479). The fluorescence antibody was PE-VEGFR2 (BD, #89106). Other antibodies included goat anti-rabbit/mouse IgG-HRP (Fdbio science, FDR007 and FDM007, Hangzhou, China) and FITC goat anti-rabbit IgG (Beyotime, A0562).

### 2.2. EN Extraction Method

EN was isolated and purified by Prof. Tian Jingkui in the Key Laboratory of Biomedical Engineering at Zhejiang University. Briefly, 10 kg of dried Radix Donggeali was heated with 60% ethanol for reflux extraction three times, and the extract was combined. Then, the extract was filtered, the solvent was concentrated until alcohol-free, and centrifuged. After centrifugation, the supernatant was separated by an HPD100 macroporous resin column, and the 30% ethanol eluent was collected. The eluent was concentrated to dry, and the total extract sample (EL) was 150 g. The EL was dissolved in 15% methanol and separated by medium-pressure preparative chromatography. The first batch was collected every 10 min. TLC detection was combined with the same fluid. A total of 20 components (EL1~20) were obtained.

### 2.3. Cell Culture

All cancer cell lines were obtained from American Type Culture Collection (ATCC, Manassas, VA, USA). HeLa cells stably expressing GFP-LC3 and L929 cells stably expressing RFP-GFP-LC3 were kindly provided by Prof. Shen Han-Ming (National University of Singapore, Singapore).

A549 and PC9 cells were maintained in RPMI 1640 medium (Genom, #GNM31800, Hangzhou, China) containing 10% fetal bovine serum (Sangon Biotech, #E510008, Shanghai, China). HUVECs were maintained in ECM medium (ScienCell, #1001, Carlsbad, CA, USA). All other cells were maintained in DMEM medium (Genom, #GNM12800) containing 10% fetal bovine serum. All cell lines were incubated at 37 °C in a 5% CO_2_ incubator (Thermo Scientific, Waltham, MA, USA).

### 2.4. Cell Viability Assay

MTT was used to measure cell viability. First, cells were seeded in 96-well plates with 5000 cells/well and incubated at 37 °C for 24 h before treatment with different concentrations of EN for different times. Then, 50 μL of MTT (2.5 mg/mL dissolved in PBS) solution was added into each well for a 2 h incubation. Finally, the medium was carefully replaced with 200 μL of DMSO. After shaking for 3 min, we measured the absorbance at 570 nm with a multiscan spectrophotometer (Thermo Scientific). The relative cell viability rate = (the average absorbance of the experimental group/the average absorbance of the control group) × 100%. All data were repeated three times.

Cell viability assay was also evaluated using a CCK8 assay kit. After the indicated treatment, 10 μL of CCK-8 was added to each well, and the cells were subsequently incubated at 37 °C for 1~4 h. The absorbance was measured at 450 nm using the microplate reader.

### 2.5. Colony Formation Assay

Cells were seeded in 6-well plates (500 cells/well) and incubated for 24 h before treatment with different concentrations of EN. The medium was removed after 24 h, and 2 mL of complete medium was added in each well. The cells were maintained for about 14 days until the clones grew to a suitable size (more than 50 cells/clone). Meanwhile, the medium was changed every three to four days. Finally, the medium was removed, and a moderate amount of PBS was used to wash each well. Methanol was used to fix the cells for 5 min, and cells were stained with 0.1% crystal violet for 30 min and then gently washed with ddH_2_O to remove excess stain. The clones were photographed and counted by Image J.

### 2.6. Tube Formation Assay

Matrigel (BD Biosciences, San Jose, CA, USA) solution was added to the well of a pre-chilled 96-well sterile plate, which was incubated for 30 min to 1 h at 37 °C to allow the Matrigel solution to form a gel. Approximately 2 × 10^4^ HUVECs/well were treated in a 96-well plate with different concentrations of EN in the presence or absence of VEGF (50 ng/mL). Then, the tube formation in each well was monitored and imaged using an inverted microscope. We used an Image J plug-in named Angiopoiesis Analyzer to analyze the tube formation. Three independent experiments were required for each treatment.

### 2.7. Transwell Migration Assay

The Transwell migration assay used six Transwell chambers (14111, Labselect, Hefei, China). Cells were seeded in an upper Transwell chamber with serum-free DMEM, and fetal bovine serum-containing DMEM containing different concentrations of EN were placed in the lower Transwell chamber. After 48 h, cells that remained on the upper surfaces of the Transwell chambers were gently removed with a cotton swab, and the cells were fixed with 4% paraformaldehyde for 15 min and stained with a crystal violet solution for 15 min. The images were measured by Image J.

### 2.8. Flow Cytometry Assay 

Phyoglobinin (PE) absorbs light at different wavelengths and can be used as a fluorescent marker. In this study, VEGFR2 expression was observed after EN treatment with HCT116 using a PE-VEGFR2 (BD, #89106) antibody. PE-VEGFR2 diluent was added and incubated in darkness at 4 °C for 2 h. Flow cytometry (Beckman, Brea, CA, USA) was used to detect the fluorescence intensity at an excitation maximum (Ex Max) of 488 nm and an emission maximum (Em Max, Kanagawa, Japan) of 576 nm.

### 2.9. Intracellular ROS Level Measurement

Cells were seeded into a 12-well plate (2 × 10^5^ cells/well) first. The next day, we replaced the medium and treated the cells for 24 h in the presence or absence of EN. DCFH-DA (2 μM) was used to treat the cells at 37 °C for 30 min. Then, cells were collected, and the fluorescence intensity was measured by flow cytometry (Beckman coulter, Brea, CA, USA). The ROS level = (experimental group/control group) × 100%. 

### 2.10. Cellular GSH Level Measurement

GSH, which is an indicator of oxidative stress, is a potential cause of apoptosis or cell death. Cells were seeded into 12-well plates (2 × 10^5^ cells/well) and then treated with EN. Finally, a GSH (10 nM) stain was added into the medium for a 30 min incubation. The cells were collected, and the fluorescence intensity was measured by flow cytometry. The GSH level = (experimental group/control group) × 100%.

### 2.11. Wound Healing

First, we drew a straight horizontal line on the bottom of the 6-well plate. Cells were seeded in the 6-well plate (5 × 10^5^ cells/well) and incubated for one or two days until 100% confluent monolayer growth. The cells were treated with different concentrations of EN for 24 h. We used a 200 μL micropipette tip to create a vertical wound and washed the exfoliated cells with PBS twice. The medium was replaced with FBS (2.5%). Finally, a suitable field was chosen for each well and photographed with INTENSILIGHT C-HGFIE (Nikon, Tokyo, Japan). The images were measured by Image J. 

### 2.12. Confocal Microscope Assay 

HeLa cells stably expressing GFP-LC3 or L929 cells stably expressing RFP-GFP-LC3 were seeded in an 8-well chamber. After the pretreatment with rapamycin (200 nM, 2 h), the cells were then treated with EN. The cell fluorescence was detected with a confocal microscope. 

HCT116 cells were first transfected with Flag-Beclin-1 and then seeded in an 8-well chamber and treated with EN. After that, the cells were fixed, permeabilized, and stained with anti-VEGFR2 conjugated to PE and LC3. Finally, Hoechst was used to stain DNA and nuclei for 10 min, and cells were photographed with a confocal microscope (LEICA TCS SP8, Leika, Wetzlar, Germany). 

### 2.13. Western Blotting 

Cells were seeded into a 6-well plate (5 × 10^5^ cells/well) and later treated with different concentrations of EN for different treatments. After that, cells were harvested and lysed in RIPA buffer (50 mM Tris-HCl (pH 7.4), 150 mM NaCl, 1% Triton X-100, 1% sodium deoxycholate, 0.1% SDS, sodium orthovanadate, sodium fluoride, EDTA, and leupeptin) with a protease inhibitor cocktail (Beyotime, #P1010). The same amount of protein was resolved by SDS-polyacrylamide gels and then transferred onto a polyvinylidene fluoride (PVDF, Bio-Rad, 1620184, Hercules, CA, USA) membrane. The membranes were blocked with TBST plus 5% non-fat milk for 2 h and incubated with primary antibodies and secondary antibodies. After the membranes were enhanced with an ECL system, they were developed with the ChemiDoc MP Imaging System (BIO-RDA).

### 2.14. In Vivo Study

Female Balb/c nude mice (3–4 weeks old) were purchased from the Shanghai SLAC Laboratory Animal Co. Ltd. (Shanghai, China). Animal welfare was ensured, and the experimental procedures strictly followed the Guide for the Care and Use of Laboratory Animals. All efforts were made to minimize animal suffering and to reduce the number of animals used. Xenograft tumors were established by subcutaneously injecting 1 × 10^7^ SW620 cells in PBS in a total volume of 0.1 mL. After one week, tumor-bearing mice were randomly divided into two groups: the vehicle group and the EN (10 mg/kg) group. The vehicle was 10% DMSO, 20% polyethylene glycol, and 5% Tween-80 in PBS. All treatments were administered by intraperitoneal injection every two days. The mice were sacrificed after another thirteen days. The tumor volume was calculated by the formula: Tumor volume = Width × Width × Length × 0.5.

### 2.15. H&E Staining Analysis

Tumors collected from mice were fixed in 4% paraformaldehyde. The paraffin-embedded samples were cut to a 4 μm thickness and stained with H&E by Wuhan servicebio technology CO. (Wuhan, China) Stained sections were viewed and photographed under a microscope.

### 2.16. Immunohistochemistry

After the mice were sacrificed, tumor tissue was removed and immediately fixed in 4% paraformaldehyde for 24 h. Immunohistochemistry was performed on tumor tissue sections to detect Ki-67 and p62 levels by Wuhan servicebio technology CO.

### 2.17. Statistical Analysis

All the experiments were performed at least three times to ensure reproducibility. Differences among the groups were analyzed by one-way variance, and the means of two groups were compared using Student’s t test with IBM Statistics SPSS 22. The results were expressed as the means ± standard errors, * *p* < 0.05, ** *p* < 0.01. As long as the *p*-value < 0.05, the difference was considered statistically significant.

## 3. Results

### 3.1. EN Inhibits Human Colon Cancer Cell Proliferation

EN is known to inhibit the activity of a variety of cancer cells [12]. Here, we investigated the cytotoxicity of EN in an array of cancer cells, including human colon cancer cells, lung cancer cells, breast cancer cells, and pancreatic cancer cells. All of these cells were exposed to EN at different concentrations (0, 8, 16, 24, and 32 μM) for 24 h, 48 h, and 72 h. We calculated the IC_50_ values of EN in eight cell lines at the time point of 48 h, based on the rate of cell inhibition (Figure 1B), and the IC_50_ values of EN in HCT116, SW620, and SW480 were 20.9 μM, 23.6 μM, and 35.8 μM, respectively. The MTT assay results also showed a dose- and time-dependent decrease in cell activity, and the cell activity curves of HCT116 and SW620 under EN treatment are shown in Figure 1C. As shown in Appendix A, we determined the effect of EN on HEK293 cell growth, which was examined by a CCK8 assay. EN exhibited little adverse effect on non-cancerous cells with its dose increasing, but it can be adjusted to a safe dose in application. In order to observe the changes in cell morphology, we used a microscope to photograph the cells under different concentrations of EN treatment. As expected, the number of cells in the visual field decreased and cells shrank, became round, and detached from the plate with an increase in the EN concentration (Figure 1D). In addition, the cell colony formation assay was also performed to observe cell activity. The number and size of the cell mass decreased in a concentration-dependent manner in HCT116 and SW620 cells (Figure 1E,F). All the above results showed that EN inhibits HCT116 and SW620 cell proliferation.

**Figure 1 molecules-27-04398-f001:**
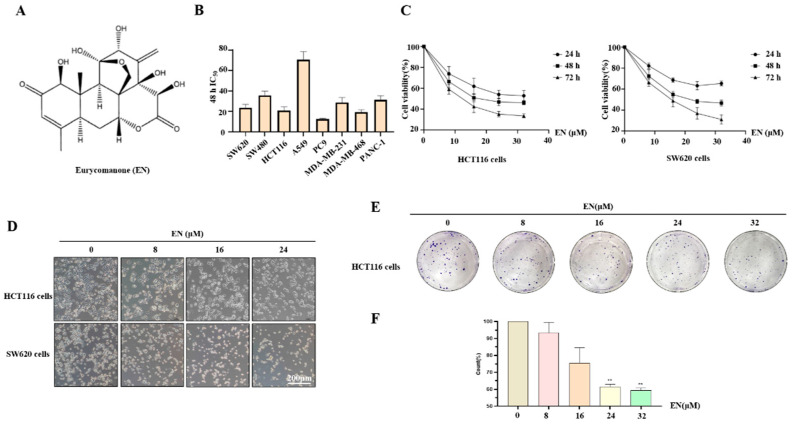
EN inhibits human colon cancer cell proliferation. (**A**) The structure of EN. (**B**) Different cancer cells were treated with EN (0, 8, 16, 24, and 32 μM) for 24 h, 48 h, and 72 h, respectively, including SW620, HCT116, SW480, A549, PC9, MDA-MB-231, MDA-MB-468, and PANC-1 cells. Cell viability was measured by MTT assay, and the 48 h IC_50_ values of EN in different cancer cells were calculated by SPSS 22. (**C**) The viability of HCT116 and SW620 cells was shown under EN treatment for 24 h, 48 h, and 72 h. (**D**) Cells were first seeded into a 12-well plate and then treated with different concentrations of EN (0, 8, 16, and 24 μM) for 24 h. The morphology and distribution of cells were photographed by microscope. (**E**) HCT116 cells were seeded into a 6-well plate and then treated with EN (32 μM) for 24 h. When the cell clusters grew to the appropriate size, cells were stained and photographed. (**F**) Image J was used to calculate the number of clones, and it was statistically analyzed by SPSS22. ** *p* < 0.01.

### 3.2. EN Inhibits Angiogenesis in Human Colon Cancer Cells

Angiogenesis has been closely associated with tumorigenesis [24]. To see whether EN also had the ability to inhibit angiogenesis, we first examined the effect of EN on human umbilical vein endothelial cells (HUVECs). The MTT assay was used to measure cell viability. In HUVECs, the cytotoxicity of EN was shown to occur in a dose- and time-dependent manner (Figure 2A). In the presence or absence of VEGF, the IC_50_ values of EN in HUVECs at 48 h were different, and they were 55.57 μM and 45.11 μM, respectively. Matrigel, a solubilized basement membrane preparation, can induce HUVECs to form blood vessels on it. We further observed the inhibitory effect of EN on HUEVC tube formation in a 96-well plate precoated with Matrigel. The branch number and total branch length showed that the number and density of new blood vessels were concentration-dependently decreased (Figure 2B,C), indicating that EN inhibits the VEGF-enhanced tube formation ability. Moreover, the VEGF-enhanced migration in HUVECs was also suppressed by an increase in the EN concentration (Figure 2D). Meanwhile, the cell number of migration was significantly decreased (Figure 2E). In addition, Western blotting was also performed to detect the expression levels of VEGFR2, and it was observed that EN downregulated the phosphorylation level of VEGFR2 in HUVEC and HCT116 cells (Figure 2F). Meanwhile, we used VEGFR2 conjugated to PE to label EN-treated HCT116 cells, and the fluorescence intensity was significantly decreased by flow cytometry (Figure 2G). The above findings confirmed that EN inhibits angiogenesis in human colon cancer cells.

### 3.3. EN Inhibits Autophagy in Human Colon Cancer Cells

To clarify the effect of EN on autophagy, Western blotting was performed to detect the LC3 expression level after EN treatment. LC3 is the marker of autophagosomes, which represents the level of autophagy. As shown in Figure 3A, the EN treatment resulted in a significant decrease in LC3-II in both HCT116 and SW620 cells. Meanwhile, the autophagy flux level was also examined using the autophagy inhibitor CQ, which inhibits autophagosome–lysosome fusion [29]. The EN treatment also decreased the level of LC-II in the presence of CQ (Figure 3A), indicating a reduced autophagic flux level. In addition, we also observed the formation of GFP-LC3 puncta under EN treatment. The rapamycin treatment significantly increased the number of GFP-LC3 puncta, but the enhancement of autophagosome formation was attenuated by the addition of EN (Figure 3B). We confirmed that EN decreases the level of autophagy. Moreover, the effect of EN treatment on the fusion of autophagosomes and lysosomes was also determined. mRFP-GFP tandem fluorescent-tagged LC3 (tfLC3) stably expressing cells were treated with EN in the presence or absence of rapamycin. As shown in Figure 3C, the EN treatment significantly attenuated the increased ratio of RFP/GFP-LC3 puncta by rapamycin, indicating the weakened formation of autolysosomes.

In human colon cancer cells, a similar result was also detected. HCT116 and SW620 cells were first pretreated with rapamycin or EBSS to activate autophagy, and then EN was added to both cells. As shown in Figure 3D, the EN treatment decreased the enhancement of autophagy either by rapamycin or EBSS starvation. In addition to the pharmacologic induction of autophagy, the genetic induction of autophagy was also performed through Beclin-1 overexpression. In HCT116 cells, Beclin-1 overexpression increased the autophagy level, but the EN treatment attenuated the upregulation of autophagy (Figure 3E).

### 3.4. EN Inhibits Autophagy through Activating the mTOR Pathway

Among the autophagy-related signaling pathways, mTOR kinase is a key regulatory molecule. The study of mTOR dates back to the discovery of rapamycin, which is a secondary metabolite secreted by soil Streptomyces [30]. In order to reveal the mechanism of EN-mediated autophagy inhibition, we examined the expression of upstream and downstream proteins of the mTOR pathway. In HCT116 cells, after EN treatment, the phosphorylation levels of ERK, mTOR, and S6 proteins were elevated, and the level of TSC2 was decreased (Figure 4A, Appendix A), indicating the activation of the mTOR signaling pathway. With increases in treatment time and dose, the activity of mTOR was further enhanced in a time- and dose-dependent manner. Similar results were also observed in another human colon cancer cell, SW620 cells (Figure 4B, Appendix A). We speculated that autophagy inhibition by EN may be attributed to the activation of the mTOR signaling pathway. 

To clarify the role of the mTOR pathway in the autophagy inhibition by EN, HCT116 cells were treated with EN in the presence of rapamycin. Rapamycin is a targeted drug for mTOR in mammals and is often used as an inducer of autophagy because it can block signals required for cell growth and proliferation to mimic cell starvation [31]. As expected, under rapamycin treatment, EN failed to activate mTOR activity and induce autophagy (Figure 4C,D, Appendix A), confirming that the mTOR pathway is a key mediator in autophagy inhibition. Similar results were also observed in SW620 cells. In addition, we also determined the role of autophagy inhibition in the anti-angiogenesis of EN. As shown in Figure 4C,D, rapamycin treatment activated autophagy. However, under the EN treatment, the upregulation of VEGFR2 was significantly attenuated (Figure 4E,F), suggesting that EN may exert its anti-angiogenesis effect by inhibiting autophagy. 

### 3.5. Autophagy Protects against EN-Caused Cell Death

In this part, we sought to find out the functional role of autophagy in EN-induced cell death. Through the genetic induction of autophagy, Flag-Beclin-1 was transfected into HCT116 cells to induce autophagy. Beclin-1 is one of the key proteins in the process of autophagic protein degradation [32]. We first conducted an MTT assay to detect the effect of Beclin-1 overexpression on cell proliferation. Beclin-1 overexpression significantly increased cell viability under the EN treatment (Figure 5A), suggesting that autophagy induction plays a protective role. In addition, we also determined the cellular redox status in EN-treated cells. In HCT116 cells, EN decreased the intracellular ROS level, while it increased the GSH level (Figure 5B). With Beclin-1 overexpression, the cellular ROS level increased, and the EN treatment attenuated the upregulation of the ROS level and alleviated oxidative stress (Figure 5C), indicating that EN possesses an anti-oxidant function. 

### 3.6. EN Inhibits the Tumorigenesis of Colon Cancer and Autophagy In Vivo

To confirm the in vitro study results, we used a xenograft model to determine the tumor suppression activity of EN. No mice died during the experiment, and no significant difference in body weight between the vehicle and EN treatment groups was observed. The tumor-bearing mice were sacrificed 24 h after the last administration of EN, and the tumor mass was removed. As shown in Figure 5D, treatment with 10 mg/kg EN inhibited tumor xenograft growth. The tumor weight decreased by 58.8% in the EN treatment group when compared with the vehicle (Figure 5E,F). These results demonstrate that EN exerts a tumor-suppressive effect in vivo.

In addition, we also examined the autophagy level of the tumor tissue. The Western blotting results showed that the levels of autophagy substrates p62 and phosphorylated S6 were upregulated after the EN treatment (Figure 5G), indicating the activation of mTOR signaling and a decrease in the autophagy level. Meanwhile, H&E staining was performed to detect the nuclear condensation of tumor tissues, and immunohistochemical staining was also performed to detect the Ki-67 and p62 protein levels. Ki-67 is present during the active phases of the cell cycle (G1, S, and G2 phase) but is absent in resting cells (G0) [33]. Thus, the high expression of Ki-67 represents the high proliferation rate of tumor cells. Our results show that the level of Ki-67-positive cells (brown) was lower in the EN treatment group than in the vehicle group (Figure 5H). On the contrary, the p62 level increased with EN treatment and was consistent with the in vitro study results. The above results demonstrate that EN inhibited the tumorigenesis of colon cancer and autophagy in vivo.

## 4. Discussion

EN is extracted from the root bark of *Eurycoma longifolia* Jack and is able to inhibit the growth of a variety of cancer cells [34]. In this study, we investigated the inhibitory effect of EN on colon cancer cell proliferation and angiogenesis and revealed the anti-cancer mechanism of EN through inhibiting autophagy (Figure 5I). 

Because pathways are not simply single-threaded, they have cross-influences, and they respond to stimuli in a timely manner. The mechanism of autophagy is far more complex than thought [35]. Here, the mTOR signaling pathway is the most important pathway in regulating autophagy. Under normal conditions, mTOR is activated by nutrients; under the conditions of nutrient and growth factor deprivation, amino acid deficiency, or low cellular energy levels, mTOR is inhibited and autophagy is activated [36]. In our study, EN treatment resulted in autophagy inhibition, which was attributed to the activation of the mTOR signaling pathway (Figure 4). It was accompanied by the increased phosphorylation levels of upstream or downstream molecules of mTOR kinase in human colon cancer cells, such as ERK, S6, and mTOR itself. Similar results were also detected in an in vivo study (Figure 5G,H). When mTOR was inhibited by rapamycin or EBSS starvation, EN failed to decrease the autophagy level of colon cancer cells, confirming the importance of mTOR in mediating autophagy. In addition to the inhibition of autophagosome formation, EN-mediated mTOR activation also influenced the fusion of autophagosomes and lysosomes. mTORC1 has been proven to be the key point in the completion of autophagy flux [37,38]. In EN-treated cells, we found that EN reduced the number of autolysosomes, with or without rapamycin treatment (Figure 3C). 

It is known that ROS is a crucial regulator of cell homeostasis in various pathways. Some stimuli that induce ROS generation can regulate autophagy as well [39,40]. ROS induces autophagy by the upregulation of Beclin-1, the oxidation of ATG4, and by causing mitochondrial dysfunction. In our study, Beclin-1 overexpression activated autophagy and resulted in an increase in cellular ROS levels (Figure 5C). Under EN treatment, autophagy was inhibited, and ROS levels also decreased. In addition, the changes in ROS level also affected cell apoptosis [41]. Excess cellular ROS have been proven to be able to activate cell apoptosis. As an antioxidant, GSH depletion is an early hallmark observed in apoptosis [42]. However, in EN-treated cells, the cellular ROS level was decreased, while the GSH level was increased (Figure 5B), indicating that EN has antioxidant activity. It was not consistent with the effect of the ROS level on the positive regulation of apoptosis. We guess that EN-mediated autophagy inhibition may regulate cell apoptosis through other pathways instead of the ROS level. However, much work is still needed in the future. 

VEGFR is a receptor of VEGF, which is an important positive factor promoting angiogenesis. Our results demonstrated the positive regulation of autophagy in angiogenesis (Figure 4D). When autophagy was activated by rapamycin, cellular VEGFR2 expression was also increased. It was reported that autophagy promotes angiogenesis via the AMPK-mTOR signaling pathway [43]. However, under treatment with anti-angiogenesis agents, the expression levels of VEGFR2 and ERK were decreased, and ERK is the positive downstream target of VEGFR2 [44,45]. In our study, VEGF was added with a concentration of 50 ng/mL in HUVECs, and VEGFR2 was decreased in response to the EN treatment (Figure 2E). Under the EN treatment, the phosphorylation level of ERK increased at first and then began to decline with time (Figure 4A,B). We guessed that the phosphorylation of VEGFR2 may happen very fast and, with the treatment time increasing, its activity decreased, which led to a reduction in ERK activity.

Based on our results, autophagy improves cell survival in the EN-treated colon cancer cells (Figure 5). Thus, we can improve the anti-colon-cancer efficacy of EN through autophagy inhibition. In a future study, we plan to combine EN with other autophagy inhibitors and investigate their synergistic effect in the treatment of colon cancer. For example, CQ, an autophagy inhibitor, not only has the ability to suppress cancer cell proliferation [46] but also to enhance the efficacy of other anti-cancer drugs [47]. In addition, other autophagy inhibitors are also available in the application of cancer therapy, such as spautin-1 [48]. 

Taken together, our results demonstrated that EN inhibits autophagy by activating the mTOR signaling pathway. In EN-treated cells, autophagy improves cell survival, which may be associated with its positive regulation of angiogenesis and cellular oxidative stress. We can make a bold prediction that EN will exert a more potent anti-tumor effect in combination with other autophagy inhibitors, angiogenesis inhibitors, or ROS inhibitors, which provides the possibility for the development of traditional medicine in cancer therapy.

## Figures and Tables

**Figure 2 molecules-27-04398-f002:**
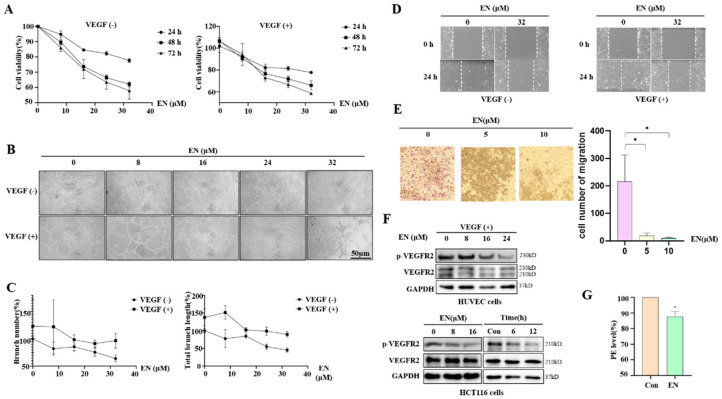
EN inhibits HUVEC and HCT116 cell angiogenesis. (**A**) HUVECs were treated with EN (0, 8, 16, 24, and 32 μM) for 24 h, 48 h, and 72 h with or without VEGF (50 ng/mL). Cell viability was measured by MTT assay. (**B**) Matrigel (30 μL) was precoated on the bottom of the 96−well plates, and 100 μL of cell suspension and 100 μL of EN diluent were added on the top. After incubating at 37 °C for 9 h, the tube was photographed with a microscope. Scale bar: 50 μm. (**C**) Image J was used to calculate the number of branches and the total length of the branches of the blood vessels. (**D**) HUVECs were seeded into the 6−well plates and then treated with EN (32 μM) with or without VEGF (50 ng/mL) for 24 h. After creating a vertical cell wound, cell culture medium was replaced, and migration level was recorded by taking photos of the same location every few hours. Scale bar: 200 μm. (**E**) Transwell migration assay. HCT116 cells were treated with EN (0, 5, and 10 μM) for 48 h. Cells that remained on the upper surfaces of the Transwell chambers were removed, and then the cells were fixed and stained. Cells were photographed and counted by image J. * *p* < 0.05. (**F**) HUVECs were treated with different concentrations of EN with VEGF (50 ng/mL) for 24 h. HCT116 cells were treated with different concentrations of EN (0, 8, and 16 μM) and for different times (0, 6, and 12 h) with EN (16 μM). Cells were harvested for Western blotting to evaluate the expression levels of VEGFR2. (**G**) HCT116 cells were first treated with EN (16 μM) for 24 h, and then cells were harvested, fixed, and permeabilized. After VEGFR2 conjugated to PE staining, flow cytometry was performed to determine cellular fluorescence intensity. * *p* < 0.05.

**Figure 3 molecules-27-04398-f003:**
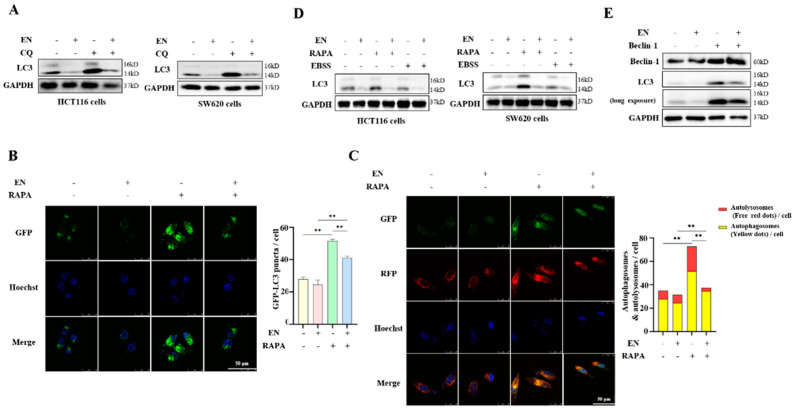
EN inhibits autophagy in colon cancer cells. (**A**) CQ (10 μM) and EN (16 μM) were used to treat HCT116 or SW620 cells for 12 h. The cells were then harvested, and Western blotting analysis was performed. (**B**) HeLa cells with stable expression of GFP−LC3 were seeded in the chambers and pretreated with rapamycin (200 nM) for 2 h. After that, cells continued to be treated with EN (16 μM) for 12 h. A confocal microscope was used to photograph the formation of GFP−LC3 puncta. Scale bar: 25 μm. Image J was used to calculate the number of GFP−LC3 puncta. ** *p* < 0.01. (**C**) As in (**B**), L929 cells with stable expression of RFP−GFP−LC3 were treated with EN in the presence or absence of rapamycin. A confocal microscope was used to photograph the formation of autolysosomes. Scale bar: 25 μm. Image J was used to calculate the number of autophagosomes and autolysosomes. ** *p* < 0.01. (**D**) HCT116 and SW620 cells were pretreated with rapamycin or EBSS for 2 h and then continued to be treated with EN (16 μM) for 24 h. Western blotting was used to detect LC3 levels. (**E**) HCT116 cells were first transfected with Flag−Beclin−1 and then treated with EN (16 μM) for 24 h. Western blotting analysis was used to detect the LC3 level.

**Figure 4 molecules-27-04398-f004:**
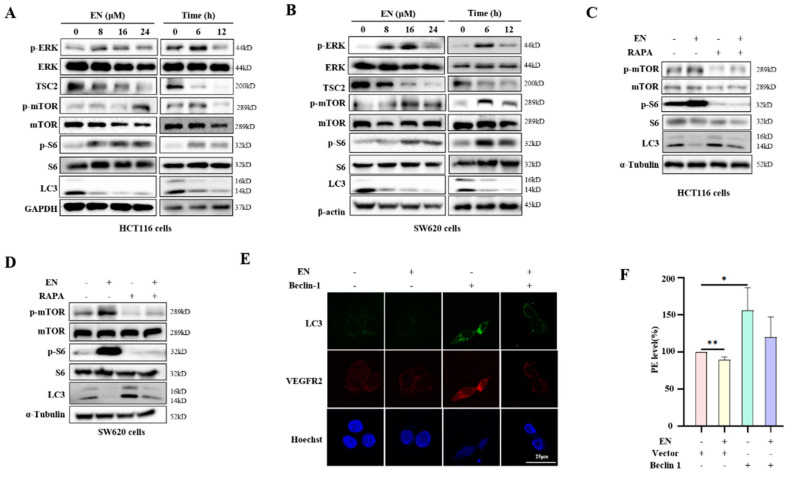
EN activates the mTOR signaling pathway and inhibits autophagy. (**A**) HCT116 cells were treated with different concentrations of EN (0, 8, 16, and 24 μM) or with EN (16 μM) for different times (0, 6, 12, and 24 h). Western blotting was used to analyze the phosphorylation levels of ERK, mTOR, and S6. (**B**) As in (**A**), SW620 cells were treated with EN and harvested for Western blotting analysis. (**C**) HCT116 cells were pretreated with rapamycin (200 nM) for 2 h and then continued to be treated with EN (16 μM) for 24 h. Cells were harvested and lysed for Western blotting to detect the expression levels of mTOR, S6, and LC3. (**D**) As in (**C**), SW620 cells were pretreated with rapamycin and then continued to be treated with EN. Cells were harvested for Western blotting analysis. (**E**) HCT116 cells were first transfected with Flag-Beclin-1 for 24 h and then treated with EN (16 μM) for 24 h. After fixation and permeabilization, cells were incubated with VEGFR2 conjugated to PE, and a confocal microscope was used to detect cell fluorescence. Scale bar: 25 μm. (**F**) As in (**D**), cell fluorescence intensity was quantified by flow cytometry and was statistically analyzed. * *p* < 0.05, ** *p* < 0.01.

**Figure 5 molecules-27-04398-f005:**
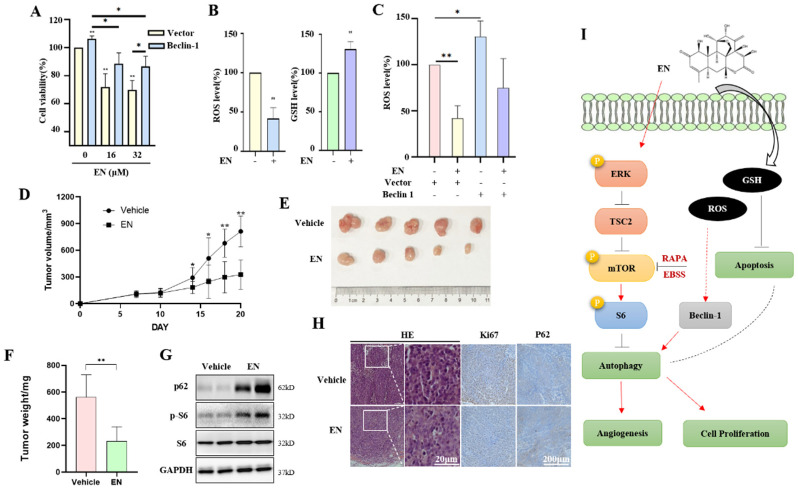
Autophagy protects against EN-induced cell death. (**A**) HCT116 cells were first transfected with Flag-Beclin-1 and then treated with EN (16 or 32 μM) for 24 h. Cell activity was measured by MTT assay. * *p* < 0.05, ** *p* < 0.01. (**B**) HCT116 cells were treated with EN (16 μM) for 24 h, and then DCFH-DA (2 μM) or GSH (10 nM) were used for cell staining. Flow cytometry was used to measure cell fluorescence. ** *p* < 0.01. (**C**) As in (**A**), EN-treated cells were stained with DCFH-DA (2 μM), and flow cytometry was used to detect cell fluorescence intensity. * *p* < 0.05, ** *p* < 0.01. (**D**) Female Balb/c nude mice were subcutaneously injected with SW620 cells to establish a xenograft tumor model, and then tumor-bearing mice were treated with EN (10 mg/kg) every two days. Tumor diameter was recorded, and tumor growth curves were plotted. * *p* < 0.05, ** *p* < 0.01. (**E**) At the end of the experiment, the mice were sacrificed, and the xenograft tumors were removed and photographed. (**F**) The excised tumors were weighted and statistically analyzed. ** *p* < 0.01. (**G**) Part of the tumor tissue was crushed and lysed for Western blotting. The expressions of p62, phospho-S6, and S6 were detected. (**H**) H&E staining and immunohistochemical staining of Ki-67 and p62proteins in the paraffin-embedded tissues were performed. Scale bar: 200 µm. (**I**) An illustrative model of autophagy inhibition in EN against colon cancer.

## Data Availability

Data available on request from the authors.

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
