# Peer review of "A Quassinoid Diterpenoid Eurycomanone from Eurycoma longifolia Jack Exerts Anti-Cancer Effect through Autophagy Inhibition"

_molecules, 2022, doi:10.3390/molecules27144398_

Round 1

Reviewer 1 Report

The manuscript submitted by Guiqin Ye et. al. describes the “A Quassinoid Diterpenoid Eurycomanone from Eurycoma Longifolia Jack Exerts Anti-cancer Effect through Autophagy Inhibition”. In this manuscript, the authors examined the regulatory effect of EN on autophagy to reveal the mechanism of EN-mediated colon cancer growth inhibition and successfully demonstrated the anti-tumor efficacy of EN in colon cancer models. Also, proved that the autophagy suppression was characterized by the downregulation of LC3-â…¡ level and the formation of GFP-LC3 puncta under EN treatment in colon cancer. Though the previous reports suggested that the Eurycomanone has an anti-tumor effect through different mechanisms in various cancers and it has been less studied in colon cancer.  I appreciate the concept and methodology of this work to prove the antitumor efficacy of Eurycomanone in colon cancer. The overall study was well designed, and adequate experiments were conducted both in vitro and in vivo and the results were very impressive. The data was well presented and explained in an understanding way. Therefore, I recommend the editor accept the manuscript with minor revision.

1)      In the Introduction authors described that “Here, our group extracted the components of Eurycoma Longifolia Jack and further improved the extraction and purification process of EN (Figure 1A). There is no protocol was mentioned in the MS. I recommend adding the methods and purification used to afford pure EN in the MS.

2)      Authors have used different cell lines for each study not the one single cell line for all the studies. For example, in Figure 5, HCT116 cells for in vitro studies and SW620 xenografts for animal studies. Is there any specific reason? I suggest explaining it in the MS.

Author Response

Dear  reviewers:

Thank you for your comments on our manuscript.Please see the attachment of our response.

Reviewer 2 Report

The authors present a manuscript entitled “A Quassinoid Diterpenoid Eurycomanone from Eurycoma Longifolia Jack Exerts Anti-Cancer Effect through Autophagy Inhibition”. The authors tried to dissect out the underlying mechanism of the anti-cancer effect of a Quassinoid Diterpenoid.

The authors performed several experiments and demonstrated that autophagic inhibition is potentially the cause of the death of cancer cells. Overall rewriting is required to convey the results properly. Several places need to require reconstruction of the sentences. Moreover, several places need some clarifications/more technical information.

Comments

Major:

In many places Figure images and legends and text potions have discrepancies.

1.     For example, in Figure 1 E “HCT116 cells were seeded into 6-well plates and then treated with EN (32 μM) for 24 h. 242 When the cells clusters grew to the appropriate size, cells were stained and photographed.” Images are given for all different time points.

2.     Image quality is not good for Fig 1 D and E

3.     Fig1B and other figures mention the number of replicates (n=?), technical or biological?

4.     Figure D images in HCT116 not many differences between16 vs 24 uM while drastic differences in Figure E 16 vs 24 uM

5.     The section “3.2. EN inhibits the angiogenesis in human colon cancer cells”. How was this experiment performed? It was not described well in the corresponding method section “tube formation assay”. If it is an in vitro angiogenesis, do you grow the colon cancer cells over the HUVEC? What was the positive control?

6.     Need better images for Angiogenesis Figure 2 B. line 272 What is Enlargement factor: 50?

7.     Figure 2F, as described in the text VEGFR2 expression checked for HUVEC and HCT116. It is not clear which one corresponds to which cell line?

8.     Figure 2F lower panel Time (Hr) blot which concentration of EN was used?

9.     Why there is no proper band of VGEFR2 in Time (Hr) blot? Same problem for upper blot. Need better blots.  Please indicate the molecular weight of the target proteins along with the blots.

10.  Need full blot images of westerns in supplements. Please indicate molecular weight of the target proteins along with the blots.

11.  Author mentioned “Meanwhile, we used VEGFR2 conjugated to PE to label EN-treated HCT116 cells and the fluorescence intensity was significantly decreased by flow cytometry (Figure 2F)” Fig 2F was showing western blot data. Where is flow cytometer data?  How did you perform the experiment? How did you perform VEGFR2 conjugated to PE labeling? What was used as the control?  Mention the experiment in the method section.

12.  Line 259-261 refer figure 2E “In addition, western blotting was also performed to detect the the expression level of VEGFR2 and it was observed that EN downregulated the phosphorylation level of VEGFR2 in either HUVEC or HCT116 cells (Figure 2E).”

But in the figure 2E is not a western blot. It is some imaging. What was it? how was it correlated to the phosphorylation level of VEGFR2?

13.  Figure legend 2G described “(G) as in (F), after EN treatment, Hoechst was used to stain the nuclei and confocal microscope was performed to photograph cell fluorescence. Scale bar: 25 μm. Image J was use to analyze the fluorescence intensity. * p < 0.05.” where is the photographed image? The corresponding plot in Fig2G shows PE level (%) cells Con vs EN.

14.  Fig 4 Time(hr) blot which EN conc?

15.  For the blots estimation required.

16.  Maximum experiments were performed on HCT116 over SW620 (like fig 5 A, B, C). But tumorigenesis was performed with SW620. Why was it chosen?

17.  HCT116 cells were first transfected with Flag-Beclin-1 and then seeded in an 8-well chamber and treated with EN (16 μM, 24 h). After transfection was the EN treatment applied immediately?

Minor

1.     Describe in a scientific way “Autophagy is an ancient process preserved in all eukaryotic cells”

2.     No section number for Statistical analysis.

3.     Why there is the additional heading of figure 2/5 written over figure 2/5 images

4.     Line 260 typos “the the”

5.     Grammatical mistake in line 282. “Image J was use to analyze the fluorescence intensity. * p < 0.05. “

Author Response

Dear reviewer:

Thank you for your comments on our manuscript.Please see the attachment about our response.

Reviewer 3 Report

Thanks to the authors, the whole idea of the manuscript is quite interesting.

However; some points need to be clarified :

1-      In figure1.B, EN was not that effective on A549 lung cancer cell line, but toxicity on colon cancer cell lines was comparable to the that of breast and pancreatic cell lines. What molecular signature or pathway in A459 could be reasoned for such resistance to EN?

2-      It is highly recommended to assess EN toxicity on non-cancerous cells in order to evaluate the likely adverse effects.

3-      In line with comment 2,  immune-staining of the xenograft tumors, instead  of western blotting is recommended. This will reveal whether the cytotoxic effects of EN is mainly tumor cells or it is going to affects the non-cancerous stromal cells as well.  

Author Response

Dear editors and reviewers:

Thanks a lot for your letter and the reviewers’ comments on our manuscript entitled " A Quassinoid Diterpenoid Eurycomanone from Eurycoma Longifolia Jack Exerts Anti-cancer Effect through Autophagy Inhibition" (ID: molecules-1742710). Those comments are very helpful for revising and improving our manuscript, as well as the important guiding significance to other research. We have read the comments carefully and revised the manuscript accordingly. The revision of the manuscript was marked with yellow color and the response to the reviewers’ comments was as follows.

  1. In figure1.B, EN was not that effective on A549 lung cancer cell line, but toxicity on colon cancer cell lines was comparable to the that of breast and pancreatic cell lines. What molecular signature or pathway in A549 could be reasoned for such resistance to EN?

Answer: Thanks for your good comments. Our studies have shown that EN affected mTOR signaling pathway in colon cancer cells. For A549 cells, they have the characteristics of wild-type EGFR[1],the activity of EGFR downstream molecules such as mTOR is lower in A549 cells, which may be the reason for their weaker response to EN treatment. For HCT116 and SW620 cells, they have the characteristics of mutation RAS[2, 3], the activity of RAS downstream molecules such as mTOR is higher, which may be the reason for their stronger reactivity to EN treatment. Thus, EN was not that effective on A549 lung cancer cell line, but toxicity on HCT116 cells and SW620 cells.

  1. It is highly recommended to assess EN toxicity on non-cancerous cells in order to evaluate the likely adverse effects.

Answer: Thank you for your suggestion. As shown in the revised supplementary Figure1, we determined the effect of EN on HEK293 cell growth, which was examined by CCK8 assay. EN exhibited little adverse effect on non-cancerous cells with its dose increasing but it can be adjusted to a safe dose in application.  

  1. In line with comment 2,  immune-staining of the xenograft tumors, instead  of western blotting is recommended. This will reveal whether the cytotoxic effects of EN is mainly tumor cells or it is going to affects the non-cancerous stromal cells as well.  

Answer: We gratefully appreciate your valuable comment. Here, we performed H&E staining analysis of tumor tissues to mark cancerous cells. As shown in the revised Figure 5H, EN treatment resulted in the downregulation of Ki67 expression levels and the upregulation of autophagy substrate p62 protein levels in cancerous cells of tumor tissue.

References

1. Choi, E.J., et al., Targeting epidermal growth factor receptor-associated signaling pathways in non-small cell lung cancer cells: implication in radiation response. Mol Cancer Res, 2010. 8(7): p. 1027-36.

2. Wong, C.C., et al., SLC25A22 Promotes Proliferation and Survival of Colorectal Cancer Cells With KRAS Mutations and Xenograft Tumor Progression in Mice via Intracellular Synthesis of Aspartate. Gastroenterology, 2016. 151(5): p. 945-960.e6.

3. Chen, P., et al., Combinative treatment of β-elemene and cetuximab is sensitive to KRAS mutant colorectal cancer cells by inducing ferroptosis and inhibiting epithelial-mesenchymal transformation. Theranostics, 2020. 10(11): p. 5107-5119.

This manuscript is a resubmission of an earlier submission. The following is a list of the peer review reports and author responses from that submission.